# Knowledge, Attitudes and Practices of Indian Immigrants in Australia towards Oral Cancer and Their Perceived Role of General Practitioners: A Cross-Sectional Study

**DOI:** 10.3390/ijerph19148596

**Published:** 2022-07-14

**Authors:** Nidhi Saraswat, Bronwyn Everett, Rona Pillay, Neeta Prabhu, Amy Villarosa, Ajesh George

**Affiliations:** 1Australian Centre for Integration of Oral Health (ACIOH), School of Nursing and Midwifery, Western Sydney University, Southwestern Sydney Local Health District, Ingham Institute for Applied Medical Research, Liverpool, NSW 1871, Australia; b.everett@westernsydney.edu.au (B.E.); amy.villarosa@westernsydney.edu.au (A.V.); a.george@westernsydney.edu.au (A.G.); 2School of Nursing and Midwifery, Western Sydney University, Parramatta, NSW 2116, Australia; rona.pillay@westernsydney.edu.au; 3School of Dentistry, Faculty of Medicine and Health, The University of Sydney, Camperdown, NSW 2006, Australia; neeta.prabhu@sydney.edu.au; 4Paediatric Dentistry, Westmead Centre for Oral Health, Sydney, NSW 2145, Australia

**Keywords:** oral cancer, knowledge, attitudes, practices, perceptions, general practitioners, Indian immigrants, Australia

## Abstract

Oral cancer is highly prevalent in the Indian subcontinent. With the increasing immigration of Indians to Australia, a potential rise in oral cancer cases can be expected if they continue engaging in oral cancer risk practices. Unfortunately, little is known on this topic in the Australian context. This study aimed to generate new insights into this area by examining Indian immigrants’ knowledge, attitudes and practices regarding oral cancer in Australia and their perceived role of general practitioners in raising oral cancer awareness. Exploring these aspects could determine the oral cancer risk behaviours of Indians in Australia along with any contributing factors which could help identify potential preventative strategies. A cross-sectional survey was undertaken of 164 Indians across Australia. Data were analysed using SPSS software with descriptive statistics. Respondents had varying levels of knowledge about oral cancer (mean total score 61%), particularly around risk factors such as alcohol and areca nut use as well as oral cancer-related signs/symptoms. The majority (87.7%) had not received any information about oral cancer in a health care setting but were receptive (71–90%) to general practitioners playing a more active role in this area. Respondents were engaging in positive preventative oral health care though few were currently (6.7%) or previously (14.7%) chewing tobacco preparations. Further research is needed through larger studies to confirm the study findings and inform the development of culturally tailored strategies particularly involving general practitioners, to raise oral cancer awareness and provide early screening for Indian immigrants.

## 1. Introduction

Oral cancer is a worldwide concern with an estimated 377,713 new cases and 177,757 deaths in the year 2020 alone [1]. The global five-year survival rate for this malignancy is still around 50% despite medical advances in the diagnosis and treatment of oral cancer [2]. Although the early identification of this disease has a critical role in improving overall survival rates, almost half of the oral cancer cases are not diagnosed until advanced stages [3]. Late detection is attributed to various factors including lack of oral cancer awareness, delay in seeking treatment from healthcare professionals and limited focus on at-risk groups [4]. One such at-risk group is the South Asian community comprising of people from countries such as India, Pakistan, Bangladesh and Sri Lanka [5]. In India, oral cancer is a leading cause of cancer-related mortality, ranking among the top cancers [6] and accounting for more than 30% of all cancers reported [7]. It is estimated there were 135,929 new lip and oral cavity cancer cases with 75,290 deaths in India for the year 2020 [8]. Thus, India is considered an epicentre of oral cancer due to the record number of cases every year [9].

The high prevalence of oral cancer among Indians is mainly attributed to the extensive consumption of tobacco products including bidis, smokeless tobacco and areca (betel) nut preparations [5]. The areca nut is the dried seed of Areca catechu and is often mistakenly termed betel nut since it is commonly chewed along with the Piper betel leaf [10]. India is the largest consumer of areca nut where the nut is cut into small pieces and chewed on its own or wrapped in a “betel vine leaf” commonly known as betel quid/‘paan’ along with slaked lime and condiments for extra flavouring [10]. Chronic use of areca nut (with or without tobacco) among this community is linked with religious beliefs, cultural acceptability, addiction and perceived advantages [5,10]. Indian migrants are well known for sustaining practices of areca nut use and betel quid (‘pan’) chewing, since these habits are deeply rooted and well-accepted customs in the Indian subcontinent [11]. Given India is the largest source of immigrants [12] and oral cancer is widespread in this country [9], the elevated risk with associated habits is also suspected to be carried by Indians to different countries [5]. The possible link between oral cancer and cultural risk practices of Indian immigrants has been researched in developed countries including the United Kingdom (UK) [13,14], the United States (US) [15], and Italy [16] with a view to promoting awareness among these populations. A recent review of this evidence concluded that South Asian immigrants including Indians in developed countries, have a lack of knowledge about oral cancer risks and are inclined towards negative oral cancer risk practices such as areca nut/betel quid chewing [17].

Over the past decade, like many developed countries, Australia has become a major multicultural ‘immigration nation’ and witnessed a significant rise in immigration from India [18,19]. Indian immigrants have had a marked surge [20] and are one of the fastest-growing communities constituting over 2.6% of the total population in Australia [21]. More recently, India was identified as the leading source with 25,698 immigrants (approximately 18.3% of all Australian immigration) [22]. Coincidently, during the period of increased immigration from India, there have also been fluctuations in the number of oral cancer cases in Australia [23]. Over the period 1997 to 2008, an increase in cancers of the tongue has been observed [23]. Likewise, in recent years, the number of lip and oral cavity cancers has increased in the continent and were projected to increase to 2788 in the year 2020 [8]. This rise in oral cancer cases could be linked to the growth of Indians in Australia [24] and their accompanying habits such as tobacco and areca (betel) nut chewing [25,26].

The potential connection between cultural risk practices of Indian immigrants and rising oral cancer has been under-researched in Australia [27,28,29,30]. However, a recent qualitative study exploring the perceptions of Indians towards oral cancer in Australia highlighted their engagement in risk habits and poor knowledge regarding adverse health effects of areca nut use [31]. This emphasizes the importance of appropriate help-seeking behaviours (HSB) which can vary with available healthcare resources as well as patient demographics and thus a balance between self-care and professional care is necessary [32]. Trained health professionals, for example, general practitioners (GPs) can play a key role in promoting health-seeking behaviours in at-risk population groups [33,34,35,36]. Nonetheless, a study exploring the perspectives and clinical practices of general practitioners (GPs) in this area revealed the need to raise awareness about new evolving oral cancer risk factors such as betel quid use among at-risk populations such as Indian immigrants [37]. Considering these emerging findings, it is important to conduct further research to shed more light on this important area of cancer care in the Australian healthcare system.

The primary aim of this study is to explore the knowledge, attitudes and practices of Indian immigrants in Australia towards oral cancer and their perceived role of general practitioners. This study was part of a broader mixed-methods study that also explored the oral cancer-related knowledge and clinical practices of GPs in Australia, especially among high-risk populations such as Indian immigrants [37]. This research was informed by both the knowledge, attitude and behaviour (K-A-B) model and the health belief model (HBM). A similar type of integrated behaviour model has been used in other international studies as well [38,39]. The KAB model, also known as knowledge, attitudes and practices (K-A-P) is an important model of health education and asserts that a behaviour change is influenced by knowledge as well as attitude [40]. Similarly, the HBM has been widely used to explain the association between attitudes and preventive health behaviours [40]. This model considers the vulnerability of the individual combined with belief that prevention is possible and can lead to actions to reduce risk. In addition, instilling positive attitudes into individuals through external avenues such as educators/health professionals—(cues to action) is likely to change their choice of action [40]. Thus, adapting both these models allowed for the perspectives and practices of both the Indian immigrants and GPs around oral cancer to be captured within the same research. The findings from this study will help identify any additional oral cancer risk behaviours in the Indian community and inform the development of culturally appropriate interventions and preventative strategies such as raising oral cancer awareness and providing oral cancer screening through GPs for early detection. This study was guided by the following research questions:What are the perceived oral health status and self-reported mouth/dental problems among Indians in Australia?What are the oral cancer-related knowledge, attitudes and practices of Indian immigrants in Australia?What are the general health and dental care practices of Indians in Australia?What are the perceptions of Indian immigrants towards general practitioners providing advice regarding oral cancer?

## 2. Materials and Methods

### 2.1. Design

A cross-sectional survey was conducted between December 2020 and February 2022 among Indian immigrants living in Australia.

### 2.2. Inclusion/Exclusion Criteria

Any person who self-identified as an Indian immigrant (were born or their parents were born in India), aged 16 years and above, and residing in Australia was eligible to participate in the survey. The minimum age of participation was set at 16 years as previous literature has shown that oral cancer risk practices such as betel nut/quid chewing are prevalent even among the younger generation in India [41]. The survey was open to Indians living in all states of Australia to ensure diversity. No restrictions were applied in terms of age limit, gender, religion, socio-economic status, occupation, residency status, place of origin in India or number of years living in Australia as international evidence in this area has revealed oral cancer risk practices to be widespread across all these variables [17].

### 2.3. Sample and Setting

The sample size was estimated on the basis of the expected proportion of the population that would engage in typical oral cancer risk practices (e.g., betel nut/quid use). Based on international published data (Indians in this context) [13,14,15,16,42] (range 34–69%), we predicted the proportion to be around 20% in Australia. A power calculation software [43] was employed to calculate the population proportion of approximately 20% with 95% confidence while allowing for a 5% margin of error. A total of 246 participants were required. Assuming roughly 80% response rate [14,15,16], 308 participants needed to be recruited.

Unfortunately, due to the coronavirus pandemic (COVID-19), the avenues of recruitment for this study were limited. The pandemic led to the implementation of physical distancing measures, national lockdowns, and travel restrictions in order to control the spread of the virus [44]. Taking these restrictions into account the survey recruitment strategy was modified from a face-to-face data collection method to a remote data collection [44]. Remote data collection refers to the collection of data via the phone and online platforms, with researchers and study participants physically distanced [45]. Therefore, all attempts were made to recruit a convenience sample [46] through online social networking sites and other virtual platforms.

The recruitment flyer for the study was published online through a webpage (link: https://cohortaustralia.com/oralcancerinimmigrants/ (accessed on 10 January 2022)) including links to the participant information sheet and the survey. The contact details of the principal researcher were provided in the flyer so that interested participants could reach out for any clarification. The study was advertised through various social media platforms including WhatsApp™ (Facebook Inc., Mountain View, CA, USA), Facebook (Facebook Inc., Menlo Park, CA, USA), and Twitter (Twitter Inc., San Francisco, CA, USA), etc. In addition, Indian associations in Australia such as the United Indian Association and the Australian Hindi Indian Association were emailed for assistance in promoting the study. Word of mouth and snowballing sampling were also used to aid recruitment.

### 2.4. Data Collection

An anonymous web-based questionnaire in English was made available through a webpage (link: https://cohortaustralia.com/oralcancerinimmigrants/ (accessed on 10 January 2022)). The survey was created using QualtricsXM (Experience management company, Seattle, WA, USA). Qualtrics is a software platform that offers a web-based survey tool for conducting survey research, evaluations, and other data collection activities. These surveys were electronically answered, and the answers were linked to the institutional QualtricsXM account. The survey required approximately 10–15 min to complete.

### 2.5. The Questionnaire Development and Pilot Testing

The study questionnaire was initially developed using existing survey items identified from a comprehensive literature review [17,47] as well as preliminary exploratory work around oral cancer in Australia [31,37]. The questionnaire development was informed by the KAB model and Health belief model. The questionnaire consisted of 39 questions which were grouped into six domains that sought information on the participants’ perceived oral health status, knowledge about oral cancer and associated risk factors, attitudes toward oral cancer risks, oral cancer risk-related practices, access to general health and oral health care, and demographic characteristics. These domains addressed various constructs of the KAB model (knowledge, attitude and practice) and HBM model (perceived benefits, perceived susceptibility, perceived barriers and cues to action) and have been detailed in Table 1. The survey items were presented as a combination of multiple-choice, Likert scales and open-ended questions. To establish content validity, the preliminary draft of the survey questionnaire was reviewed by an expert panel consisting of academic and clinical experts in the field of oral cancer, dentistry, nursing and public health (*n* = 5). The comments from the panel were taken into consideration and minor revision of items was undertaken. Thereafter, the revised survey was pilot tested with seven Indians in Australia for face validity. These participants were asked for feedback on the readability and clarity of the questions and the duration of the survey. The survey was then modified according to their suggestions, and the final version of the survey was used for data collection through Qualtrics. The survey tool/questionnaire has been attached as Appendix A.

### 2.6. Measures

The measures which formed parts of the survey questionnaire and data analysis are listed below. Standardised questions that were validated to assess oral health status and oral cancer risk behaviours were used where available.

### 2.7. Data Analysis

Data were analysed through Statistical Package for the Social Sciences (SPSS) Version 27 software [v.27, IBM, New York, NY, USA]. Socio-demographic variables, health-specific characteristics, self-reported oral health status, and oral cancer-related knowledge, attitudes and behaviours were summarised using descriptive statistics. This included mean and standard deviation for continuous variables and frequency counts and percentage for categorical variables. Pearson’s chi-square tests were used to test for associations between categorical variables, including oral cancer-related behaviours and attitudes. Group differences in continuous variables, such as knowledge, were either assessed using independent samples T-tests if normally distributed, or otherwise Mann–Whitney U tests. The significance level for all analyses was set at *p* < 0.05.

### 2.8. Ethical Considerations

This study received ethics approval from the Human Research Ethics Committee of Western Sydney University (H13203). No incentive was offered to participants. The survey was online and completely anonymously. Participation was voluntary and submission of a completed questionnaire implied consent to participate. All responses were recorded in an online database and accessible by the research team only.

## 3. Definition of Terms

The terms ‘knowledge’, ‘attitudes’ and ‘practices’ have been used widely in this paper. For the purposes of this paper, the definition of ‘knowledge’ refers to one’s level of information, awareness and understanding relating to oral cancer risk [17]. The term ‘Attitudes’ has been employed to depict the individuals’ views, inclinations, perceptions, and beliefs associated with oral cancer risk [17]. The ‘practices’, in the current context, relates to oral cancer risk-related habits and the actions regarding initiation, continuation or quitting of these habits [17]. The reference to ‘risk products’ has been provided to depict potential oral cancer causative products such as tobacco and areca (betel) nut preparations [50]. ‘Immigrant’ term, in this paper, refers to a person who moves into a country other than that of his/her nationality [51].

## 4. Results

### 4.1. Demographic Characteristics

A convenience sample of 164 Indian immigrants was recruited across Australia. A total of 192 Indians accessed the survey. Of these, 28 participants were excluded as they completed less than 50% of the questions offered to all respondents, leaving a total of 164 cases included in the analysis. Slightly less than half (47.2%) of the respondents were female, and the age ranged from 20 to 69 years, with an average age of 35.2 years. Participants had lived in India for between 2 to 60 years, and over three-quarters (87.1%) of the respondents spoke a language other than English at home. The majority (90%) had attended university, and more than half (69.3%) were working full time. Full demographic characteristics of the sample can be seen in Table 2.

### 4.2. Perceived Oral Health Status

Nearly two-thirds (65.8%, *n* = 102) of the respondents rated their oral health as excellent or good and almost a quarter (23.8%, *n* = 39) currently had problems or concerns with their oral health.

### 4.3. Oral Cancer-Related Knowledge

Around three-quarters (73.8%, *n* = 121) of the respondents had heard about oral cancer and the mean total knowledge score was 12.8 out of a total of 21 possible points (SD4.06) with a range of 5–20. The lowest numbers of correct responses were seen in items regarding pain from oral cancer screening and the capacity of GPs to perform them (24.4–45.7%), symptoms of oral cancer such as painless ulcers red patches yellow patches discomfort and bleeding gums (14–48.8%), and family history as a risk factor for oral cancer (41.8%, *n* = 64). Participants were comparatively less knowledgeable about betel quid/nut and alcohol as risk factors for oral cancer compared to traditional factors such as smoking or chewing tobacco. Higher mean knowledge scores were reported among respondents who had heard about oral cancer compared to those who had not heard about oral cancer (13.1 vs. 11.2; Mann–Whitney U = 1595.5, *p* = 0.012) as well as among those reporting a family history of oral cancer compared to those who had no family history of oral cancer (76.9% vs. 47.2% Pearson chi-square = 4.155, 1df, *p* = 0.042). All knowledge items and number of correct responses are presented in Table 3.

### 4.4. Oral Cancer-Related Attitudes

When asked to rate the importance of various activities in the prevention of oral cancer, more respondents rated a healthy diet (64.1%), brushing teeth twice daily (73.2%), and regular dental visits (65.4%) as very important. See Table 4 for ratings of all activities.

Almost half (45.8%, *n* = 70) of the respondents thought people of Indian background were at higher risk of oral cancer. When asked why they thought people used tobacco preparations and alcohol, the largest proportion of respondents indicated this was due to addiction (83.0%, *n* = 127) and leisure/lifestyle/enjoyment (69.3%, *n* = 106), and the smallest proportion of responses were regarding it being a cultural practice for some Indians (32.7%, *n* = 50).

Over half of the respondents indicated they would visit a dentist for a white or coloured patch in the mouth (55.6%, *n* = 85) that had lasted more than 3 weeks and a doctor for an ulcer or sore in the mouth that had lasted more than 3 weeks (56.2%, *n* = 86). More than half (60.1%, *n* = 92) of respondents indicated they prefer seeing health professionals from their cultural background for regular check-ups.

### 4.5. Oral Cancer-Related Practices

Just under half (42.4%) of the respondents indicated consuming drinks that contain alcohol. Few reported currently smoking (6%) or chewing tobacco preparations (6.7%). Between 14 and 18% of participants reported previously smoking and chewing tobacco preparations (see Table 5 for full results).

Smoking and chewing tobacco preparations were significantly associated with gender, whereby being male was associated with higher proportions of currently smoking (*p* = 0.003), ever smoking (*p* < 0.001), currently chewing (*p* = 0.006) and ever chewing (*p* < 0.001) tobacco preparations. See Table 6 for all associations.

### 4.6. Engagement in Health and Dental Care Services

Approximately three-quarters (72.8%) of the respondents reported having visited a doctor, community health clinic or practice nurse in the last 12 months; however, just over one-third (35.4%) had visited a dentist in the last 12 months. Only 12.3% (*n* = 18) of respondents recalled receiving information about oral cancer during any health care visit.

### 4.7. Perceived Role of General Practitioners

Roughly three-quarters of the respondents (70.5%, *n* = 103) agreed that a doctor could assist them in identifying oral health problems such as oral cancer. While just half of the respondents (50.0%, *n* = 73) thought doctors had sufficient knowledge to advise them regarding oral cancer, 71.9% (*n* = 105) would consider oral health advice given by their doctor and 90.4% (*n* = 132) would make an appointment to see a dentist if referred by a doctor.

## 5. Discussion

This descriptive cross-sectional study is the first to examine the oral cancer-related knowledge, attitudes and practices of Indian immigrants in Australia and their perceptions of the role of general practitioners in this area. The study sample had diverse socio-demographic characteristics which were fairly representative of the Australian population data for Indian immigrants in terms of their median age (35.2 vs. 33.9 years) and proportion of Indian males (52.8 vs. 54.6%) [52,53]. Most respondents had a tertiary level of education and were employed full-time, which is consistent with recent immigration statistics showing Indian immigrants holding the highest number of skilled visas [52]. Similar to national data, most were followers of the Hinduism religion and were residing in either NSW or VIC as these two states are favoured by most Indian-born immigrants [53].

Overall, there was varying knowledge regarding oral cancer. In line with previous research [54], majority of the respondents were knowledgeable about oral cancer risks posed by smoking (92.2%) and chewing tobacco (94.1%). Unlike previous studies that demonstrate low levels of knowledge regarding the carcinogenic effects of areca (betel) nut [13,15,16], three-quarters of the respondents in our study reported they were aware of this as a risk factor. The higher level of knowledge could be due to the fact the Indians in our study were more highly educated (close to 90% graduates) and economically well-off when compared to the past literature (85% completed high school and 69% employed) [15] as there is clear evidence showing both these factors are linked to increased oral cancer awareness [55,56]. It was perhaps unsurprising that the demographic profile of the current study sample was different to other developed countries given Australia has had a selective immigration policy focusing primarily on highly skilled migrants [57]. Interestingly, consistent with past research involving Asian males in the UK [58], not all (64.7%) in the current study knew about the association of alcohol use with oral cancer. This finding may not be unique to immigrants from South Asia as studies across UK and Australia show that the general public are equally uninformed about alcohol being a risk factor for oral cancer [56,59].

The survey findings also reaffirm an earlier report [31] that Indian immigrants in Australia have limited knowledge of the potential signs and symptoms of oral cancer. Less than half (46.3%) of the respondents believed that a white or red patch/discoloration in the mouth could be a sign of oral cancer. A previous study in the UK comprising young South Asians also reported similar findings [13]. This lack of awareness could have serious implications as neglecting any discolorations in the mouth can lead to delayed oral cancer screening and diagnosis which can be detrimental due to the high mortality rates associated with this type of cancer [60]. Further exacerbating the situation is the fact that less than a quarter of the respondents were aware that screening for oral cancer can be carried out by a general practitioner. A possible reason for this could be that promoting oral cancer awareness among patients in general healthcare settings has often been neglected across developed countries due to various barriers [61,62]. This is clearly evident in the study findings as very few (12.3%) acknowledged receiving oral cancer information during medical appointments. This lack of knowledge may have influenced the attitudes of the respondents with more than half preferring to visit a dentist over a GP for patches or any discolorations in the mouth. This could have important clinical implications in developed countries such as Australia where access to affordable dental care is limited, resulting in infrequent dental visits among immigrants as seen in the study findings.

It was also encouraging to note that many respondents acknowledged the importance of preventive oral health activities such as brushing teeth twice a day (73.2%) and regular dental visits (65.4%). These positive views could have been influenced by socio-economic characteristics (such as education level and income) and the healthy immigrant effect [63] although this does deteriorate over time. A high education level, as seen in this study, has been shown to be positively linked to health behaviours such as physical activity, brushing and diet [63]. This finding also complements the observations from another study in the USA which reported very positive attitudes towards oral health among almost 80% of Indian immigrants [64]. Nevertheless, in line with qualitative findings [31], almost half of the respondents (45.8%) agreed that Indian immigrants in Australia were at higher risk for oral cancer due to their continued use of risk products (e.g., areca nut/tobacco preparations and alcohol) and linked this to addiction, leisure/lifestyle choices and cultural customs. These views are not surprising as the burden of oral cancer in India [7,65] and the prevalence of risk practices among Indians has been well documented with research continuing to show that these practices tend to continue even after migration to new settlements [5,11,66]. This is particularly relevant in Australia with recent reports [25,26] suggesting the popularity of new risk products such as areca nut and the potential flow-on effect on future oral cancer cases.

Respondents’ knowledge about key oral cancer risk factors and their positive attitudes toward preventive oral health may have had a positive impact on their practices with less than half (42.4%) indicating they consumed alcohol and even less reported smoking (6%). In addition, very few reported chewing tobacco and areca nut preparations (6.7%) which is in sharp contrast to previous studies which have shown higher prevalence rates of these practices (range 34–69%) [13,14,15,16,42]. This finding was also dissimilar to our earlier qualitative study [31] which showed indulgence (occasional/regular) of all Indian immigrants in one or more risk habits such as smoking, alcohol and betel nut/quid use; however, it is important to note that almost 15% of the respondents had previously used tobacco and areca nut products. As we did not assess the average period since cessation of these practices, we cannot be certain that the prevalence rates reported for chewing tobacco preparations in this study are accurate. Adding to this is the fact that the current study was undertaken during the COVID-19 lockdown in Australia and as a consequence respondents may have had limited access to tobacco preparations due to restricted domestic travel, limited supply of Indian groceries and their inability to travel overseas to India [67]. A pilot study in the USA [15] did find that difficulty in procurement/storage of tobacco preparations and being socially unacceptable were some of the reasons why respondents decided to switch or stop risk practices such as betel quid use. It is also possible that the number of years living in Australia could have influenced the oral cancer risk practices of Indian migrants. The respondents had been living in Australia for a long period (average of 26 years) and this could have affected their risk habits as a result of acculturation which is usually common in South Asian migrants [68]. Due to the very small number of people using areca nut preparations in the sample, a test of associations between length of residence and areca (betel) nut use was not possible. Nevertheless, this aspect should be considered in future research. Smoking or chewing tobacco and areca (betel) nut products were found more frequent in males as compared to females. This finding is supported by research from the US [15] which showed that men were larger consumers (61%) of betel quid/’pan’ compared to women (26%). Interestingly, studies do show there is considerable use of areca nut preparations among women in India [69,70].

The study findings suggest the need for strategies to raise oral cancer awareness among this population group at the community, health services and policy levels. Although there was greater awareness around key oral cancer risk products such as smoking and tobacco chewing and few appeared to be engaging in such risk practices, the variable levels of knowledge around other risk factors such as alcohol and areca nut use as well as oral cancer-related signs/symptoms are concerning. Culturally appropriate programs and visual aids such as posters raising awareness of oral cancer risks associated with tobacco and betel quid/’pan’ use could be displayed at avenues popular for Indian gatherings such as Asian/Indian grocery stores, restaurants, temples and cultural events. Social media such as Facebook, Instagram, WhatsApp, Twitter, and local Indian radio and television channels could also assist in spreading relevant public health messages. Although no relationship was found between religion and frequency of risk habits in this study (possibly due to the small sample), previous studies have shown differences in knowledge levels and health-related behaviours among South Asian subgroups with diverse religious backgrounds [71]. More research is thus needed from an Australian context to inform the development of preventive educational campaigns/resources that are evidence based and tailored to specific subgroups within the Indian community.

Another key strategy is to involve general practitioners in oral cancer prevention and awareness programs as they can play a pivotal role in this area. They have a trusting relationship with their patients to address this issue by providing educational counselling relating to oral cancer risk and the importance of early intervention. This is supported by the fact that most respondents clearly engaged more with health services than dental care services and were very receptive to receiving oral health advice and referrals from general practitioners. However, there was a lack of clarity among respondents about the exact scope of practice of GPs and this is evident by the fact that many indicated they would see a GP for a nonhealing ulcer/sore and not a white/coloured patch in the mouth. A number of respondents also questioned the competency of GPs to advise them on oral cancer. These findings are not unforeseen as a recent qualitative study involving GPs in Australia raised similar concerns regarding their limited knowledge around new oral cancer risk factors, inconsistent clinical practices relating to routine oral cancer check-ups and referrals [37]. These views were supported by the findings of a review that explored the knowledge, attitudes and practices of general practitioners in developed countries regarding oral cancer [47]. Based on the current literature, it is evident that for GPs to take up this unique opportunity, additional training is required around emerging risk factors such as areca (betel) nut preparations, which could be provided via continuing professional development programs and online modules. One such resource targeted at health professionals in public health services has recently been launched in one state in Australia which offers great insights into the carcinogenic potential of areca (betel) nut use [72]. This is a step in the right direction, but more can be done in this field, particularly around consumer resources relating to oral cancer and betel nut which could be promoted in the waiting room of health care settings. To our knowledge, no national resource is currently available in Australia regarding areca nut use among immigrants and this is an area that needs urgent attention particularly given the ongoing influx of Indian immigrants to Australia [22].

It is important to note that many participants (60%) expressed their preference for health professionals of similar cultural backgrounds for routine medical/dental check-ups. This mirrors the observations from an earlier study which found that having GPs from similar ethnic backgrounds is preferable for Indian women due to a perceived better understanding of cultural issues and being able to communicate in the patients’ own language [73]. Past research has also found that immigrants from the Indian subcontinent do not feel comfortable visiting doctors/dentists of different ethnic backgrounds as communication may be limited or sometimes ineffective and culturally insensitive advice from the healthcare provider may offend immigrant patients hindering healthcare delivery [5]. These points need to be taken into consideration when targeting and training GPs to raise oral cancer awareness among the Indian population.

To ensure consistent clinical practices in this area, guidelines need to be developed by health departments and professional organisations to ensure all medical and dental professionals engaging with Indian immigrants provide the same messaging regarding oral cancer risk practices. Apart from this, the rising oral cancer burden draws attention towards a growing need for opportunistic oral cancer screening and effective monitoring systems in Australia [23] to assess oral cancer cases in immigrants. Lastly, it is important to note that the key focus of the current study was on hypothesis generation to shed more light on this under-researched area. The findings have identified emerging areas that need continued research through larger studies. These areas include, confirming if Indians in Australia are engaging in oral cancer risk behaviours particularly chewing tobacco preparations, examining if the oral cancer risk behaviours are different between first- and second-generation Indian immigrants and confirming the perspectives and clinical practices GPs in Australia towards oral cancer, especially when interacting with high-risk populations such as Indian immigrants.

The results of this study should be interpreted with caution due to several limitations. Firstly, the sample size was small owing to recruitment challenges during the COVID-19 pandemic and thus more complex analysis was not possible. More importantly, due to convenience sampling, the study findings may not be reflective of the whole Indian community living in Australia and therefore cannot be generalized. Further, most of the participants had been residing in Australia for a long period and so the findings may not reflect the current knowledge and practices of new immigrants. The majority of participants were also in their middle age and thus more research with younger and older generations could give further insights into this research area. In addition, the reported results are subject to information bias and due to the self-reported data as well as social desirability bias, the respondents may have under-reported their oral cancer risk practices. Despite these limitations, the survey has broadened our understanding of this under-researched topic in Australia and identified potential pathways to raise oral cancer awareness in this community.

## 6. Conclusions

This study has revealed varying levels of knowledge about oral cancer among the sample of Indian immigrants, particularly around risk factors such as alcohol and areca nut use as well as oral cancer-related signs/symptoms. Positive attitudes about preventive oral health practices were evident though some were involved in oral cancer risk practices. The findings have also highlighted the lack of adequate information regarding oral cancer being provided in primary health care settings and uncertainty around the scope of practice of GPs in this area. The receptiveness of the study sample towards GPs playing a role in raising oral cancer awareness looks promising and adequate training of these health professionals could be beneficial. With the growing influx of Indian immigrants to Australia, more strategies are needed to raise awareness in this community about oral cancer risk practices, particularly around tobacco/areca nut use which is highly prevalent in India. Further research through larger studies and a more representative sample is warranted to explore this area in Australia and confirm the study findings. Greater knowledge in this area will help inform the development of culturally sensitive and tailored strategies to raise awareness of oral cancer risk among Indian immigrants.

## Figures and Tables

**Table 1 ijerph-19-08596-t001:** Measures of questionnaire.

Domain	Description	Constructs from KAP and HBM Model
Perceived oral health status	4 items A single item question widely used in the previous studies [48,49] to assess overall oral health status (excellent, very good, good, fair and poor). A single item question to describe teeth, gum and mouth problems (yes, no), with a list of most common oral health problems/concerns found in people [17].	
Knowledge about oral cancer and associated risk factors	27 items Sourced from previous studies and included some validated items [13].	Knowledge
Attitudes toward oral cancer risks	19 items Sourced from previous studies and included some validated items [13].	Attitudes Perceived benefits Perceived susceptibility Perceived barriers
Oral cancer risk-related practices	9 items Sourced from previous studies and included some validated items [13].	Practices
Access to health care	13 items Sourced from previous studies and included some validated items [13].	Practices Cues to action
Socio-demographic questions	16 items	Modifying variables

**Table 2 ijerph-19-08596-t002:** Demographics characteristics of included participants.

Characteristic	*n* (%)
Gender ^†^	
Male	76 (52.8)
Female	68 (47.2)
Age at last birthday (mean ± SD) ^†^	35.2 ± 7.40
Years lived in India (mean ± SD) ^†^	8.2 ± 6.19
Years since living in Australia (mean ± SD) ^†^	26.1 ± 7.87
Country of birth	
Australia	2 (1.4)
India	142 (98.6)
State of residence ^‡^	
NSW	48 (47.5)
ACT	1 (1.0)
VIC	30 (29.7)
QLD	2 (2.0)
SA	15 (14.9)
WA	3 (3.0)
TAS	1 (1.0)
NT	1(1.0)
Speaks English at home ^†^	18 (12.9)
Current religion ^‡^	
Atheist/not religious	2 (2.4)
Catholic	4 (4.8)
Christian	10 (11.9)
Hindu	66 (78.6)
Islam	1 (1.2)
Jain	1 (1.2)
Level of education ^†^	
Primary school	2 (1.4)
High school	6 (4.3)
TAFE	6. (4.3)
University	126 (90.0)
Employment status ^†^	
Full time	97 (69.3)
Part time	12 (8.6)
Casual	18 (12.9)
Home/domestic duties	4 (2.9)
Retired	3 (2.1)
Not working	6 (4.3)
Average household income ^†^	
Less than AUD 40,000	14 (10.0)
AUD 40,000–59,000	13 (9.3)
AUD 60,000–79,000	15 (10.7)
AUD 80,000–99,000	12 (8.6)
AUD 100,000–120,000	13 (9.3)
More than AUD 120,000	37 (26.4)
Don’t know	1 (0.7)
Prefer not to answer	35 (25.0)
Has private health insurance ^†^	
Yes	62 (44.3)
No	70 (50.0)
Don’t know	8 (5.7)
Has family history of cancer/oral cancer ^†^	
Yes	13 (9.3)
No	114 (81.4)
Don’t know	11 (7.9)
Prefer not to answer	2 (1.4)

^†^ Number of missing cases for each item ranged from 20–39; ^‡^ number of missing cases for each item ranged from 63–80.

**Table 3 ijerph-19-08596-t003:** Knowledge items and correct responses.

Item	Correct *n* (%)
A check up for mouth (oral) cancer is:	
Painless	75 (45.7)
A way of finding mouth (oral) cancer at an early stage	104 (63.4)
Helps in treatment of oral cancer if detected early	129 (78.7)
Can be done by a GP	40 (24.4)
The signs/symptoms of mouth (oral) cancer are:	
A white patch/discoloration in the mouth	76 (46.3)
An ulcer (sore) that does not heal	95 (57.9)
A painless ulcer (sore) in the mouth	71 (43.3)
A red patch in the mouth	76 (46.3)
A yellow patch in the mouth (correct response: false)	23 (14.0)
A lump or swelling in the mouth	103 (62.8)
A sore throat	49 (29.9)
Discomfort or pain in the mouth	68 (41.5)
Bleeding gums	80 (48.8)
You are more likely to get oral cancer if you:	
Smoke tobacco, cigars or pipe	141 (92.2)
Smoke hukkah (sheesha)	133 (86.9)
Chew tobacco	144 (94.1)
Drink alcohol heavily	99 (64.7)
Chew gutkha	142 (92.8)
Chew betel quid/‘pan’	116 (75.8)
Chew betel nut/‘supari’	119 (77.8)
If your family got it	64 (41.8)

Number of missing cases per item ranged from 0–11.

**Table 4 ijerph-19-08596-t004:** Self-rated importance of various activities in the prevention of oral cancer.

Preventive Activities	Not Important *n* (%)	Slightly Important *n* (%)	Fairly Important *n* (%)	Important *n* (%)	Very Important *n* (%)
Doing exercise regularly	16 (10.5)	16 (7.8)	22 (14.4)	35 (22.9)	68 (44.4)
Eating a healthy diet (2 fruits and 5 vegetables per day)	5 (3.3)	3 (2.0)	29 (19.0)	18 (11.8)	98 (64.1)
Brushing teeth twice a day	1 (0.7)	4 (2.6)	14 (9.2)	22 (14.4)	112 (73.2)
Visit a dentist at least once a year	2 (1.3)	6 (3.9)	25 (16.3)	20 (13.1)	100 (65.4)
Visit a doctor (G.P.) regularly	2 (1.3)	10 (6.5)	30 (19.6)	37 (24.2)	74 (48.4)

Number of missing cases per item was 11.

**Table 5 ijerph-19-08596-t005:** Self-reported behaviours with alcohol and tobacco preparations.

	Yes *n* (%)	I Used to and Stopped *n* (%)	Never *n* (%)
Consume drinks that contain alcohol	64 (42.4)	21 (13.9)	66 (43.7)
Smoke tobacco (including cigarettes, cigars, pipes or hukkah)	9 (6.0)	27 (17.9)	115 (76.2)
Chew any tobacco preparations (tobacco, betel nut/supari, betel quid/pan)	10 (6.7)	22 (14.7)	118 (78.7)

Number of missing cases per item ranged from 13–14.

**Table 6 ijerph-19-08596-t006:** Associations between gender and use of tobacco preparations.

	Male *n* (%)	Female *n* (%)	Pearson’s Chi-Square (df)	*p*-Value
Currently smokes tobacco preparations	9 (11.8)	0 (0.0)	8.589 (1)	0.003
Currently chews tobacco preparations	8 (10.5)	0 (0.0)	7.579 (1)	0.006
Has ever smoked tobacco preparations	29 (38.2)	4 (5.9)	21.164 (1)	<0.001
Has ever chewed tobacco preparations	25 (32.9)	3 (28)	18.588	<0.001

## Data Availability

Data are available from the corresponding author upon reasonable request.

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
