# Peer review of "Knowledge, Attitudes and Practices of Indian Immigrants in Australia towards Oral Cancer and Their Perceived Role of General Practitioners: A Cross-Sectional Study"

_ijerph, 2022, doi:10.3390/ijerph19148596_

Round 1

Reviewer 1 Report

This manuscript reports original research based on an online questionnaire administered to an ethnic minority group in Australia. The small convenience group of participants (I would not call this a sample if the study population cannot be clearly specified), the descriptive nature of the data and the use of an old-fashioned behaviour model (K-A-B for knowledge, attitudes and practices) are the main limitations of the study. Some of those limitations were raised in the discussion, although they cannot be fully addressed. I would recommend the following:

Mention clearly among the limitations of the study that the sample is not representative of the Indian group living in Australia and therefore results cannot be generalised beyond the study group. Furthermore, rephrase the conclusion to match the study aim (explore Indian immigrants’ knowledge, attitudes and practices regarding oral cancer in Australia and their perceived role of general practitioners raising oral cancer awareness). Don’t use statements such as “This study has reaffirmed that Indian immigrants in Australia” or “The findings have also confirmed that Indian immigrants are not receiving adequate information” or “Indian migrants are engaging in positive preventative” because your sample was not representative. Don’t make it sound like you have a national sample of immigrants. Whatever you found is applicable to only this group of Indian adults.

Also, that the K-A-B model used to design the questionnaire is considered outdated and irrelevant in public health practice. Alternative behaviour change models would have shed more lights on the precursors and determinants of the relevant behaviours. I would recommend mentioning the health belief model, the transtheoretical model of change or the COM-B model as a minimum. Finally, state that this was not hypothesis-testing but a mere hypothesis-generating study in the discussion. This should be followed by at least 3 novel hypotheses stemming from this research for further research.

Finally, I would encourage the authors to run some regression models for associations. One I would be particularly interested to see is whether behaviour varies according to immigration status (first or second generation) and socioeconomic position (education, employment and income, even if you have to merge categories to create at least 3 groups given the small numbers).

Author Response

Comment 1: This manuscript reports original research based on an online questionnaire administered to an ethnic minority group in Australia. The small convenience group of participants (I would not call this a sample if the study population cannot be clearly specified), the descriptive nature of the data and the use of an old-fashioned behaviour model (K-A-B for knowledge, attitudes and practices) are the main limitations of the study. Some of those limitations were raised in the discussion, although they cannot be fully addressed. I would recommend the following:

Authors’ response: Thank you for the feedback. We have addressed the comments separately below.

Comment 1a: Mention clearly among the limitations of the study that the sample is not representative of the Indian group living in Australia and therefore results cannot be generalised beyond the study group.

Authors’ response: Thank you for the suggestion. We acknowledge that the sample was small and recruited via convenience sampling (due to limited options during pandemic- as mentioned in the paper) and agree that the findings may not be reflective of the Indian community living in Australia. We have now highlighted this point as a limitation in the paper. (Pg.13, line 512-514)

Comment 1b: Furthermore, rephrase the conclusion to match the study aim (explore Indian immigrants’ knowledge, attitudes and practices regarding oral cancer in Australia and their perceived role of general practitioners raising oral cancer awareness).

Authors’ response: Thank you for the suggestion. We have rephrased the conclusion to match the study aim, which now reads as-

This study has revealed varying levels of knowledge about oral cancer among the sample of Indian immigrants particularly around risk factors like alcohol and areca nut use as well as oral cancer-related signs/symptoms. Positive attitudes about preventive oral health practices were evident though some were involved in oral cancer risk practices. The findings have also highlighted the lack of adequate information regarding oral cancer being provided in primary health care settings and uncertainty around the scope of practice of GPs in this area. The receptiveness of the study sample towards GPs playing a role in raising oral cancer awareness looks promising and adequate training of these health professionals could be beneficial. With the growing influx of Indian immigrants to Australia, more strategies are needed to raise awareness in this community about oral cancer risk practices particularly around tobacco/areca nut use which is highly prevalent in India. Further research through larger studies and a more representative sample is warranted to explore this area in Australia and confirm the study findings. Greater knowledge in this area will help inform the development of culturally sensitive and tailored strategies to raise awareness of oral cancer risk among Indian immigrants.’

(Pg.13, line 524-536; Pg.14, line 556-557)

Comment 1c: Don’t use statements such as “This study has reaffirmed that Indian immigrants in Australia” or “The findings have also confirmed that Indian immigrants are not receiving adequate information” or “Indian migrants are engaging in positive preventative” because your sample was not representative.

Authors’ response: Thank you for the comment. As mentioned in comment 1a and 1b we acknowledge that our findings cannot be generalised to all Indians residing in Australia and have therefore revised the conclusion accordingly. (Pg.13, line 524-536; Pg.14, line 556-557)

Comment 1d: Don’t make it sound like you have a national sample of immigrants. Whatever you found is applicable to only this group of Indian adults.

Authors’ response: Thanks for the feedback. We have revised the conclusion to make it clear that the findings were only applicable to the study sample and cannot be generalised. We have also highlighted the need for larger studies with more a more representative sample to confirm the study findings (Pg.13, line 524-536; Pg.14, line 556-557).

Comment 2: Also, that the K-A-B model used to design the questionnaire is considered outdated and irrelevant in public health practice. Alternative behaviour change models would have shed more lights on the precursors and determinants of the relevant behaviours. I would recommend mentioning the health belief model, the transtheoretical model of change or the COM-B model as a minimum.

Authors’ response: Thank you for the comment. This study examined the knowledge, attitudes and practices of Indian immigrants in Australia towards oral cancer and their perceived role of general practitioners. This study was part of a broader mixed-methods study that also explored the oral cancer-related knowledge and clinical practices of GPs in Australia, especially among high-risk populations such as Indian immigrants (1). This research was informed by both the K-A-B model and the Health Belief model. A similar type of integrated behaviour model has been used in other international studies as well (2,3). Adapting both these models allowed for the perspectives and practices of both the Indian immigrants and GPs around oral cancer to be captured within the same research program. We have made this clear at the end of the introduction of the manuscript (Pg.3, line 104-120).

The questionnaire development for this study was also informed by both these models. The questionnaire domains addressed various constructs of the KAB model (knowledge, attitude and practice) and HBM model (perceived benefits, perceived susceptibility, perceived barriers and cues to action) and this has been detailed in the Methods section (Pg. 4, line 182-183/187-190) and Table 1 (Pg. 5).

  1. Saraswat, N., et al., Perceptions and Practices of General Practitioners towards Oral Cancer and Emerging Risk Factors among Indian Immigrants in Australia: A Qualitative Study. International journal of environmental research and public health, 2021. 18(21): p. 11111.
  2. Hsieh, H.-M., et al., Mediation effect of health beliefs in the relationship between health knowledge and uptake of mammography in a National Breast Cancer screening program in Taiwan. Journal of Cancer Education, 2021. 36(4): p. 832-843.
  3. Rimpeekool, W., et al., “I rarely read the label”: factors that influence Thai consumer responses to nutrition labels.Global Journal of Health Science, 2016. 8(1): p. 21.)

Comment 3: As Finally, state that this was not hypothesis-testing but a mere hypothesis-generating study in the discussion. This should be followed by at least 3 novel hypotheses stemming from this research for further research.

Authors’ response: Thank you for your comment. We have included additional information at the end of the discussion confirming that the key focus of this study was on hypothesis generation and have put forward three areas stemming from the findings that require further research. The additional section reads as follows:

‘Lastly, it is important to note that the key focus of the current study was on hypothesis generation to shed more light on this under-researched area. The findings have identified emerging areas that need continued research through larger studies. These areas include confirming if Indians in Australia are engaging in oral cancer risk behaviors particularly chewing tobacco preparations, examining if the oral cancer risk behaviors are different between first- and second-generation Indian immigrants, and confirming the perspectives and clinical practices GPs in Australia toward oral cancer, especially when interacting with high-risk populations like Indian immigrants.’  (Pg. 13, line 501-509)

 Comment 4: Finally, I would encourage the authors to run some regression models for associations. One I would be particularly interested to see is whether behaviour varies according to immigration status (first or second generation) and socioeconomic position (education, employment and income, even if you have to merge categories to create at least 3 groups given the small numbers).

Authors’ response: Thank you for this suggestion. Unfortunately, as only 2 participants indicated being born in Australia, immigration status could not be included in a regression model. A binary logistic regression model was constructed with the dependent variable of having ever chewed tobacco preparations (currently chew + used to chew vs never chewed). However, none of the included independent variables returned significant associations: average income collapsed at median (OR= 5.584, p=0.210), received information re oral cancer (OR=4.461, p=0.333), follows Hindu religion (OR=1.399, p=0.736), total knowledge score collapsed at median (OR=0.256, p=0.158), level of education (OR=0.074, p=0.174), employed full time (OR=2.796, p=0.373). As a result, this model has not been included in the paper's findings. This is in line with the included findings, which highlighted gender as the only significantly associated characteristic with chewing tobacco preparations.

Reviewer 2 Report

The authors chose a difficult field to investigate the importance of awareness in a minority of Australian citizens who come from a background with a high prevalence of oral cancer. A well-done study with a small number of participants. The questionnaire is exceptionally well-designed. The study analysis is concise. Statistics appear to be adequate. I recommend accepting.

Author Response

Comment 1: The authors chose a difficult field to investigate the importance of awareness in a minority of Australian citizens who come from a background with a high prevalence of oral cancer.

Authors’ response: Thank you for this feedback.

Comment 2: A well-done study with a small number of participants. The questionnaire is exceptionally well-designed. The study analysis is concise. Statistics appear to be adequate. I recommend accepting.

Authors’ response: Thank you for the positive comments.

Reviewer 3 Report

Abstract:

The abstract has not mentioned the reason/rational to why investigate knowledge, attitude and practitioners. Does the article attempt to inductively or deductively induce an impact on the early detection or knowledge sharing? I am not sure the argument is clearly articulated! Please provide a cause and effect argument, what and how the findings will improve oral cancer detection or treatment!

Introduction

How the identification of correlation between cultural risk practices and rising oral cancer will be informed by attitudes and perceptions in either qualitative or quantitative context.

Question 2, is this tackling the inspection or early detection visually? Or the awareness smoker subjects are more likely to develop oral cancer?

Question 3 is this within the Australian health care system or in general?

Question 4, this is independent variable which I am not sure how this will contribute to the main aim of the study?

Recruitment challenge could have been managed through online recruitment channels or social media

Author Response

Comment 1: In Abstract-

Comment 1a: The abstract has not mentioned the reason/rationale to why investigate knowledge, attitude and practices.

Authors’ response: Thank you for the comment. As suggested, the abstract has been modified to include a rationale for investigating knowledge, attitude and practices. (Pg.1, line 23-25)

Comment 1b: Does the article attempt to induce an impact inductively or deductively on the early detection or knowledge sharing?

Authors’ response: Thank you for the raising this point. This study was a hypothesis generating study since little is known on this topic (in the Australian context). We have made this clear in the abstract (Pg.1, line 32-34) and reiterated it at the end of the discussion (Pg. 13, line 501-509).

Comment 1c: I am not sure the argument is clearly articulated! Please provide a cause-and-effect argument, what and how the findings will improve oral cancer detection or treatment!

Authors’ response: We have revised the abstract to provide a better argument and rationale for the study.(Pg.1, line 32-34).

Comment 2: In Introduction-

Comment 2a: How the identification of correlation between cultural risk practices and rising oral cancer will be informed by attitudes and perceptions in either qualitative or quantitative context.

Authors’ response: Thanks for your comment. Typical cultural practices of South Asians, particularly Indians to consume areca nut in various preparations has been highlighted in several studies. Adding to this, areca nut has been identified as a risk factor with carcinogenic potential to cause oral cancer and past literature in other developed countries have highlighted the role of Indian immigrants’ perceptions (in terms of perceived benefits and religious beliefs) in continuing cultural risk practices like areca nut use even in new settlements and the implications in relation to oral cancer (1,2,3), this study aimed to explore this association in the Australian context. Furthermore, some recent reports (4,5) on areca/betel nut use have surfaced in Australia despite the prohibitions on sale and consumption of areca nut (these points have been previously explained in the introduction section in paper- paragraph 2 and 3).

To reinforce these points, we have now included additional information (at the end of the introduction) about the conceptual models that informed this study namely the knowledge, attitude and behaviour (K-A-B) model and Health Belief model (HBM). This integrated behaviour model helps show that the correlation between cultural risk practices (e.g., areca nut use) and rising oral cancer cases could be informed by attitudes and perceptions in either the qualitative (Saraswat, N., et al., Oral cancer risk behaviours of Indian immigrants in Australia: A Qualitative Study. Australian and New Zealand Journal of Public Health, 2021) or quantitative context (current study), which in turn can inform preventative strategies for oral cancer prevention. (Pg.3, line 104-120).

  1. Changrani, J., et al., Paan and gutka use in the United States: a pilot study in Bangladeshi and Indian-Gujarati immigrants in New York City. Journal of immigrant & refugee studies, 2006. 4(1): p. 99-109.
  2. Petti, S., and S. Warnakulasuriya, Betel quid chewing among adult male immigrants from the Indian subcontinent to Italy. Oral Diseases, 2018. 24(1-2): p. 44-48
  3. Auluck, A., et al., Areca nut and betel quid chewing among South Asian immigrants to Western countries and its implications for oral cancer screening. Rural and remote health, 2009. 9(2): p. 1118)
  4. Australian Dental Association. Media Release - Dentists Issue Warning on Hazards of Betel Quid Chewing. Sydney (AUST): ADA; 2020. Available from: Available from: https://www.ada.org.au/getattachment/News-Media/ News-and-Release/Media-Releases/World-Cancer-Day- 2020/03-02-20-Dentists-issue-warning-over-rising- practice-in-Australia-of-chewing-betel-nut.pdf.aspx.
  5. Faa M. Betel nut black market boom in Australia has experts warning of devastating health impacts. ABC Far North [Internet]. 2020 Available from: https://www.abc.net.au/news/2020-02- 04/cancer-causing-betel-nut-booms-on-Australian- black-market/11925148

Comment 2b: is this tackling the inspection or early detection visually? Or the awareness smoker subjects are more likely to develop oral cancer?

Authors’ response: Thanks for the comment. We have clarified this point at the end of the introduction by stating the following:

‘The findings from this study will help identify any additional oral cancer risk behaviours in the Indian community and inform the development of culturally appropriate interventions and preventative strategies like raising oral cancer awareness and providing oral cancer screening through GPs for early detection’. (Pg 3 lines 116-120)

Comment 2c: is this within the Australian health care system or in general?

Authors’ response: Thanks for your comment. We have clarified that the focus of GPs in this study is within the Australian healthcare system. (Pg.2, line 98)

Comment 2d: this is independent variable which I am not sure how this will contribute to the main aim of the study?

Authors’ response: Thanks for your comment. The primary aim of this study is to explore the knowledge, attitudes and practices of Indian immigrants in Australia towards oral cancer and we believe the reviewer is referring to the second part of the study aim namely “their perceived role of general practitioners”. If this is the case, we believe this aspect is related to the study aim as interaction with health professionals like GPs can play a key role in influencing people’s behaviours and reducing risk. This is well articulated in the Health Belief Model (referred to as Cues to action) which informed this study. (These points have been added to the last paragraph of the introduction). (Pg.3, line 104-120)  

Comment 3: Recruitment challenge could have been managed through online recruitment channels or social media.

Authors’ response: Thanks for the feedback. We acknowledge that online channels or social media are an effective way of recruiting participants to address recruitment challenges. We had opted for this method of recruitment as mentioned in the paper (Pg.4, line 159-160; Pg. 4, line 167-170). In addition, we also approached Indian associations in Australia to assist in recruitment. However, it is possible that south Asian (including Indians) populations can be hard to recruit as highlighted in past literature and this is possible why the online recruitment strategy was not that effective.

Reviewer 4 Report

Thank you for giving me to review your manuscript. This manuscript is interesting and scientifically meaningful for considering the help-seeking behaviors of immigrants in Australia. Regarding the contents, the following revision should be considered.

The title should include the study design, and should be described as cross-sectional study.

The introduction paragraph should explain theoretical and conceptual frameworks. The theoretical framework should be oral cancer and risk factors, and the conceptual framework may be the risk factors and help-seeking behaviors regarding oral cancer.

In the introduction, the relationship between help-seeking behaviors and general physicians should be delineated more depth. The authors should describe the situation of the relationship between Help-seeking behaviors and general physicians/primary care doctors by using the following articles.

- Shaw, C., et al., How people decide to seek health care: a qualitative study. Int J Nurs Stud, 2008. 45(10): p. 1516-24.

- Ohta, R., et al., Potential Help-Seeking Behaviors Associated with Better Self-Rated Health among Rural Older Patients: A Cross-Sectional Study. Int J Environ Res Public Health, 2021. 18(17).

- Ansar, A., et al., Duration of intervals in the care seeking pathway for lung cancer in Bangladesh: A journey from symptoms triggering consultation to receipt of treatment. PLoS ONE, 2021. 16(9 September): p. e0257301.

- Ohta, R., et al., The association between the self-management of mild symptoms and quality of life of elderly populations in rural communities: A cross-sectional study. International Journal of Environmental Research and Public Health, 2021. 18(16): p. 8857.

- Birhanu, Z., et al., Health seeking behavior for cervical cancer in Ethiopia: A qualitative study. International Journal for Equity in Health, 2012. 11(1): p. 83.

This study should describe the limitation of sampling bias and the results' applicability to other settings, and the future investigation in the limitation part.

In conclusion, the statement is too strong, and the authors should be modest in the statement. This study can just say the possibility with a small sample size.

Author Response

Comment 1: Thank you for giving me to review your manuscript. This manuscript is interesting and scientifically meaningful for considering the help-seeking behaviors of immigrants in Australia.

Authors’ response: Thanks for the feedback.

Comment 2: Regarding the contents, the following revision should be considered:

Authors’ response: Thanks for suggesting revisions. We have addressed comments as given below:

Comment 2a: The title should include the study design and should be described as cross-sectional study.

Authors’ response: We acknowledge that the title should reveal the study design and have modified it accordingly ‘Knowledge, Attitudes and Practices of Indian Immigrants in Australia towards oral cancer and their perceived role of General Practitioners: A cross sectional study’

Comment 2b: The introduction paragraph should explain theoretical and conceptual frameworks. The theoretical framework should be oral cancer and risk factors, and the conceptual framework may be the risk factors and help-seeking behaviors regarding oral cancer.

Authors’ response: Thank you for this suggestion. We have included in the introduction the underlying conceptual models that informed this study to better explain the various components. The following section has been added:

The primary aim of this study is to explore the knowledge, attitudes and practices of Indian immigrants in Australia towards oral cancer and their perceived role of general practitioners. This study was part of a broader mixed-methods study that also explored the oral cancer-related knowledge and clinical practices of GPs in Australia, especially among high-risk populations such as Indian immigrants [37]. This research was informed by both the knowledge, attitude and behaviour (K-A-B) model and Health Belief model (HBM). A similar type of integrated behaviour model has been used in other international studies as well [38, 39]. The KAB model, also known as knowledge, attitudes and practices (K-A-P) is an important model of health education and asserts that a behaviour change is in-fluenced by knowledge as well as attitude [40]. Similarly, HBM has been widely used to explain the association of attitudes and preventive health behaviours [40]. This model considers the vulnerability of the individual combined with belief that prevention is possible can lead to actions to reduce risk. In addition, instilling positive attitudes into individuals through external avenues like educators/health professionals – (cues to action) is likely to change their choice of action [40]. Thus, adapting both these models allowed for the perspectives and practices of both the Indian immigrants and GPs around oral cancer to be captured within the same research. The findings from this study will help identify any additional oral cancer risk behaviours in the Indian community and inform the development of culturally appropriate interventions and preventative strategies like raising oral cancer awareness and providing oral cancer screening through GPs for early detection.’ (Pg.3, line 104-120).

Comment 2c: In the introduction, the relationship between help-seeking behaviors and general physicians should be delineated more depth. The authors should describe the situation of the relationship between Help-seeking behaviors and general physicians/primary care doctors by using the following articles:

- Shaw, C., et al., How people decide to seek health care: a qualitative study. Int J Nurs Stud, 2008. 45(10): p. 1516-24.

- Ohta, R., et al., Potential Help-Seeking Behaviors Associated with Better Self-Rated Health among Rural Older Patients: A Cross-Sectional Study. Int J Environ Res Public Health, 2021. 18(17).

- Ansar, A., et al., Duration of intervals in the care seeking pathway for lung cancer in Bangladesh: A journey from symptoms triggering consultation to receipt of treatment. PLoS ONE, 2021. 16(9 September): p. e0257301.

- Ohta, R., et al., The association between the self-management of mild symptoms and quality of life of elderly populations in rural communities: A cross-sectional study. International Journal of Environmental Research and Public Health, 2021. 18(16): p. 8857.

- Birhanu, Z., et al., Health seeking behavior for cervical cancer in Ethiopia: A qualitative study. International Journal for Equity in Health, 2012. 11(1): p. 83.

Authors’ response: Thanks for the comment and suggested references. As suggested, the relationship between help-seeking behaviours and general physicians has been elaborated in brief in the paper. (Pg.2, line 89-93).

This now reads as:

‘However, a recent qualitative study exploring the perceptions of Indians towards oral cancer in Australia highlighted their engagement in risk habits and poor knowledge regarding adverse health effects of areca nut use [31]. This emphasizes the importance of appropriate help-seeking behaviours (HSB) which can vary with available healthcare resources as well as patient demographics and thus a balance between self-care and professional care is necessary [32]. The trained health professionals, for example, general practitioners (GPs) can play a key role in promoting health seeking behaviours in at-risk population groups [33-36]’.

Comment 3: This study should describe the limitation of sampling bias and the results' applicability to other settings, and the future investigation in the limitation part.

Authors’ response: Thanks for the suggestion. We acknowledge that the findings from the small sample recruited by convenience sampling (opted due to COVID 19 pandemic restrictions during this study) cannot be generalised to all the Indians living in Australia and that is a limitation. We have addressed this limitation more explicitly in paper now and have suggested that future research with larger and more representative sample recruited by random sampling would be more beneficial to explore this area in depth and to inform preventative strategies (Pg.13, line 524-536; Pg.14, line 556-557).

Comment 4: In conclusion, the statement is too strong, and the authors should be modest in the statement. This study can just say the possibility with a small sample size.

Authors’ response: Thanks for the comment. We agree with the reviewer and as suggested have now modified the conclusion section accordingly. (Pg.13, line 524-536; Pg.14, line 556-557)

Round 2

Reviewer 1 Report

The authors have addressed all of my comments in the first round of reviews. The manuscript reads well now. 

Reviewer 4 Report

The manuscript has been considerably improved. I think that this paper is suited for inclusion in our journal.